# Detection of Spinal Muscular Atrophy Patients Using Dried Saliva Spots

**DOI:** 10.3390/genes12101621

**Published:** 2021-10-14

**Authors:** Yogik Onky Silvana Wijaya, Hisahide Nishio, Emma Tabe Eko Niba, Kentaro Okamoto, Haruo Shintaku, Yasuhiro Takeshima, Toshio Saito, Masakazu Shinohara, Hiroyuki Awano

**Affiliations:** 1Department of Community Medicine and Social Healthcare Science, Kobe University Graduate School of Medicine, 7-5-1 Kusunoki-cho, Chuo-ku, Kobe 650-0017, Hyogo, Japan; yogik.onky@gmail.com (Y.O.S.W.); niba@med.kobe-u.ac.jp (E.T.E.N.); mashino@med.kobe-u.ac.jp (M.S.); 2Faculty of Rehabilitation, Kobe Gakuin University, 518 Arise, Ikawadani-cho, Nishi-ku, Kobe 651-2180, Hyogo, Japan; 3Department of Pediatrics, Ehime Prefectural Imabari Hospital, 4-5-5 Ishiicho, Imabari 794-0006, Ehime, Japan; kentaro206@gmail.com; 4Department of Pediatrics, Osaka City University Graduate School of Medicine, 1-4-3 Asahi-Machi, Abeno-ku, Osaka 545-8585, Osaka, Japan; shintakuh@med.osaka-cu.ac.jp; 5Department of Pediatrics, Hyogo College of Medicine, 1-1 Mukogawacho, Nishinomiya 663-8501, Hyogo, Japan; ytake@hyo-med.ac.jp; 6Department of Neurology, National Hospital Organization Osaka Toneyama Medical Center, 5-1-1 Toneyama, Toyonaka 560-8552, Osaka, Japan; saito.toshio.cq@mail.hosp.go.jp; 7Department of Pediatrics, Kobe University Graduate School of Medicine, 7-5-1 Kusunoki-cho, Chuo-ku, Kobe 650-0017, Hyogo, Japan; awahiro@med.kobe-u.ac.jp

**Keywords:** dried saliva spot, spinal muscular atrophy, *SMN1*, modified competitive oligonucleotide priming-polymerase chain reaction, melting peak analysis, nested PCR

## Abstract

Spinal muscular atrophy (SMA) is a lower motor neuron disease, once considered incurable. The main symptoms are muscle weakness and muscular atrophy. More than 90% of cases of SMA are caused by homozygous deletion of survival motor neuron 1 (*SMN1*). Emerging treatments, such as splicing modulation of *SMN2* and *SMN* gene replacement therapy, have improved the prognoses and motor functions of patients. However, confirmed diagnosis by *SMN1* testing is often delayed, suggesting the presence of diagnosis-delayed or undiagnosed cases. To enable patients to access the right treatments, a screening system for SMA is essential. Even so, the current newborn screening system using dried blood spots is still invasive and cumbersome. Here, we developed a completely non-invasive screening system using dried saliva spots (DSS) as an alternative DNA source to detect *SMN1* deletion. In this study, 60 DSS (40 SMA patients and 20 controls) were tested. The combination of modified competitive oligonucleotide priming-polymerase chain reaction and melting peak analysis clearly distinguished DSS samples with and without *SMN1*. In conclusion, these results suggest that our system with DSS is applicable to SMA patient detection in the real world.

## 1. Introduction

Spinal muscular atrophy (SMA) is one of the most devastating neuromuscular disorders, characterized by motor neuron degeneration [1]. The patients present with muscle weakness and muscular atrophy [1]. Clinically, SMA is divided into five subtypes: Type 0 (the most severe form, prenatal onset, death within weeks of birth); Type I (Werdnig–Hoffmann disease, severe form, onset < 6 months, non-sitter); Type II (Dubowitz disease, intermediate form, onset < 18 months, sitter); Type III (Kugelberg–Welander disease, mild form, onset > 18 months, walker); and Type IV (the mildest form, onset > 30 years) [2,3].

The majority of SMA cases (~95%) involve homozygous deletion of survival motor neuron 1 (*SMN1*), while some others (~5%) carry a deleterious *SMN1* mutation, thus *SMN1* is considered the disease-causing gene [1,2]. The homologous gene, *SMN2*, serves as an SMA-modifying gene because a high *SMN2* copy number is generally associated with a milder phenotype [4,5].

*SMN1* and *SMN2* are identical except for five nucleotides located in intron 6, exon 7, intron 7, and exon 8 [1]. Molecular diagnosis of SMA requires the detection of *SMN1*-specific nucleotides, especially the one located in exon 7. Several methods have been developed to achieve this, including single-stranded conformation polymorphism (SSCP) analysis [1], restriction enzyme digestion analysis [6], modified competitive oligonucleotide priming-polymerase chain reaction (mCOP-PCR) [7,8], multiplex ligation-dependent probe amplification (MLPA) [9], digital droplet PCR (ddPCR) [10] and next-generation sequencing (NGS) [11].

SMA has long been thought an incurable disease because of the absence of effective drugs. However, three new and effective drugs for SMA are now available: nusinersen [12], onasemnogene abeparvovec-xioi [13], and risdiplam [14]. These drugs give better outcomes when treatment is initiated at an early stage, before the onset of symptoms [12,13,14]. Therefore, there is a growing implementation of newborn screening programs for SMA (NBS-SMA) worldwide [7,15,16,17,18,19,20,21]. Technically, the current NBS-SMA program can detect all SMA patients with a homozygous deletion regardless of the clinical subtype.

Even though NBS-SMA has begun in many countries, there remain many older children or adults with SMA in the population who were not detected in early infancy. The majority of them may be diagnosed with SMA in a timely fashion, but the rest may be diagnosed late or may not be diagnosed, particularly those with a later onset of the disease [22,23,24,25]. Undiagnosed patients cannot access treatment with the new drugs. Therefore, to complement NBS-SMA, it is desirable to establish a screening system that covers older children and adults. The samples for that purpose should be collected at schools, workplaces, or homes. However, one potential hurdle is the invasiveness of blood-based templates such as dried blood spots (DBS), which may require health care visitation of the individuals or blood collection by health care workers.

Here, we aim to develop a non-invasive SMA screening method from dried saliva spots (DSS) to detect *SMN1* using a combination of nested mCOP-PCR and melting curve analysis. The simplicity and non-invasiveness of the DSS collection might be suitable for adults and children school-aged or older. To our knowledge, there has been no reported study employing DSS as genetic material to screen for SMA.

## 2. Materials and Methods

### 2.1. Patient and Control Samples

A total of 61 DSS samples (40 SMA patients and 21 controls) were enrolled in this study. The patients had been diagnosed as having *SMN1*-deleted SMA based on an *SMN1* deletion test using the PCR restriction fragment length polymorphism (PCR-RFLP) method [6].

Prior to this study, informed consent was obtained from all participants. The study was approved by the Ethics Committee of Kobe University Graduate School of Medicine (reference number 1210, approved on 12 October 2011) and was conducted in accordance with the World Medical Association Declaration of Helsinki.

### 2.2. Detection of SMN1 and CFTR

#### 2.2.1. Outline of the Detection System

Our system consisted of two steps, multiplex nested PCR amplification and melting curve analysis. (1) Multiplex nested PCR. The first-round PCR amplified *SMN1* exon 7 and *SMN2* exon 7 using a common primer set, and simultaneously amplified *CFTR* as a reference gene. The second-round PCR amplified *SMN1* exon 7 with a specific mCOP primer set and simultaneously amplified *CFTR* as a reference gene. (2) Melting curve analysis. The dissociation characteristics of the nested PCR products during heating were analyzed. In this analysis, the *SMN1* peak was clearly separated from the *CFTR* peak.

#### 2.2.2. Preparation of DSS Samples

The chemically modified filter paper was used in this study, Indicating Flinders Technology Associates (FTA) card (GE Healthcare, Boston, MA, USA), which is designed to inactivate highly pathogenic organisms. The DSS sample was prepared by direct spitting into the center of the FTA card and was then air-dried for at least one hour. The DSS cards were sent to our laboratory for immediate analysis or stored in a dark room at room temperature until use.

#### 2.2.3. Multiplex Nested PCR

##### First-Round PCR: Multiplex Amplification of SMN and CFTR Outer Fragments

A conventional multiplex PCR was used to amplify outer fragments of *SMN1/SMN2* and *CFTR* using the GeneAmp® PCR System 9700 (Applied Biosytems, Foster City, CA, USA).

The DSS was punched out in the center of the spot area using a 2 mm diameter paper punch. The punched paper was then placed into a 50 μL PCR mixture for direct amplification containing 1 U of DNA polymerase KOD FX Neo (TOYOBO, Osaka, Japan). A blank paper was also punched out and used as a negative control. The primers used to amplify the outer fragment of *SMN**1* and *SMN2* were R111 (5′-AGA CTA TCA ACT TAA TTT CTG ATC A-3′) and 541C770 (5′-TAA GGA ATG TGA GCA CCT TCC TTC-3′) (Figure 1A) [1]. The primers used to amplify the outer fragment of *CFTR* were CF621F (5′-AGT CAC CAA AGC AGT ACA GC-3′) and CF621R (5′-GGG CCT GTG CAA GGA AGT GTTA-3′) (Figure 1A) [26].

The PCR conditions were: (1) initial denaturation at 94 °C for 7 min; (2) 35 cycles of denaturation at 94 °C for 30 s, annealing at 60 °C for 30 s, and extension at 72 °C for 30 s; (3) additional extension at 72 °C for 7 min; and (4) hold at 10 °C.

To confirm the amplification of the target sequences, the first-round PCR products were run on a 4% agarose gel in 1 × TBE buffer and visualized by Midori-Green staining (Nippon Genetics, Tokyo, Japan). The first-round PCR products were then diluted 100-fold and used as the template for the second-round PCR.

##### Second-Round PCR: Allele-Specific Amplification of SMN1 and CFTR Inner Fragments

A real-time mCOP-PCR was performed to specifically amplify the inner fragment of *SMN1* using a LightCycler® 96 system (Roche Applied Science, Mannheim, Germany). Here, the inner fragment of *CFTR* was also amplified, together with *SMN1*, as a reference gene.

Two microliters of a 100-fold dilution of the first-round PCR product were added into a reaction mixture with a final volume of 30 µL containing 1 U of DNA polymerase KOD FX Neo and 1.5 µL EvaGreen^®^ Dye (Biotium, Hayward, CA, USA). The primers for allele-specific amplification of the *SMN1* inner fragment were RIII and SMN1-COP (5′-TGT CTG AAA CC-3′) (Figure 1B) [1,27]. The primers for amplification of the *CFTR* inner fragment were CF621F2 (5′-ATC ATA GCT TCC TAT GAC CCG GA-3′) and CFTR-COP (5′-GGC TGG GTG TA-3′) (Figure 1B).

The PCR conditions were as follows: (1) initial denaturation at 94 °C for 7min; (2) 20 cycles of denaturation at 94 °C for 30 s, annealing at 37 °C for 30 s, and extension at 72 °C for 30 s; and (3) melting curve analysis consisting of an initial holding step at 65 °C for 1 min, a heating step with a temperature increase from 65 to 97 °C at a rate of 0.2 °C/s, and a final holding step at room temperature.

To confirm the amplification of the target sequences, the second-round PCR products were run on a 4% agarose gel in 1 × TBE buffer and visualized by Midori-Green staining.

#### 2.2.4. Melting Curve Analysis

Melting curve analysis was performed with LightCycler^®^ 96 Software (version 1.1.0.1320). The temperature conditions are described above. Fluorescence data were converted into melting peaks by the software and plotted as the negative derivative of fluorescence with respect to temperature (−dF[fluorescence]/dT[temperature] vs temperature) [28]. To quantify the melting peak profiles, we calculated the *SMN1*/*CFTR* ratio (SCR). The SCR values are defined as the *SMN1* melting peak height divided by the *CFTR* melting peak height (see the Results section).

### 2.3. Statistical Analysis

To compare differences in the SCR between the two groups, the Student’s *t*-test was performed using Microsoft Excel with the add-in software Statcel 3 (The Publisher OMS Ltd., Tokyo, Japan). A *p*-value of less than 0.05 was considered statistically significant.

## 3. Results

### 3.1. Preliminary Assays

We analyzed three fresh samples using conventional PCR and gel electrophoresis: a control DSS sample, a patient DSS sample, and a blank filter paper sample as a negative control. Nested PCR products of these samples were electrophoresed on an agarose gel (Figure 2A). The control DSS sample showed two bands for the *SMN1* (169 bp) and *CFTR* (111 bp) products, whereas the patient DSS sample showed only one band for *CFTR*, indicating the absence of *SMN1* amplification. *CFTR* has been often used as the reference gene [26,28,29].

Next, instead of using conventional PCR analysis and agarose gel electrophoresis, we analyzed the samples using real-time PCR and melting curve analysis. To detect the amplification products of *CFTR* and *SMN1*, melting curve analysis was performed immediately after real-time PCR (Figure 2B). The control DSS sample showed two melting peaks for the *SMN1* (77 °C) and *CFTR* (82 °C) products, whereas the patient DSS sample showed only one peak for *CFTR*, indicating the absence of *SMN1* amplification.

Then, we also checked the possibility of DNA degradation in the DSS sample during the storage period. We analyzed the same control DSS sample 1, 2, 3, and 4 months after saliva collection. Two melting peaks for the *SMN1* and *CFTR* products were observed in all the experiments performed at 1, 2, 3, and 4 months.

Here, we defined the SCR value using the peak height of those two genes. The value tended to decrease as the months increased, suggesting that *SMN1* degraded more easily than *CFTR* (Figure 3). However, we also noticed that the SCR value did not obviously decrease within one month after saliva collection. Thus, we analyzed all DSS samples within two weeks in the main experiments.

### 3.2. Main Assays

#### 3.2.1. Melting Curve Analysis of 60 DSS Samples

We collected DSS from 21 healthy controls and 40 SMA patients. These SMA patients enrolled in this study had been confirmed to carry no *SMN1* exon 7 using the PCR-RFLP method [6].

All the samples were successfully analyzed except for one DSS sample of a healthy control (1 out of 21 DSS). This sample showed no amplification in the first-round PCR (data not shown). This negative result might be due to the absence of DNA or the presence of PCR inhibitors. We then excluded this sample from further analysis. In total, we analyzed 20 DSS of healthy controls and 40 DSS of SMA patients.

All 20 DSS samples from healthy controls produced two melting peaks for *SMN1* and *CFTR*, whereas all 40 DSS samples from SMA patients had only one melting peak for *CFTR*. The melting peak profiles of the patients using DSS were consistent with their RFLP-PCR results using freshly collected blood.

#### 3.2.2. SCR Values between Controls and SMA Patients

In the early stage of the development of our system, the results of melting curve analyses were only examined by visual inspection by two individuals. However, to properly deal with poor-quality or poor-quantity DNA samples, we needed to quantify the melting peak pattern and determine the cutoff point of our assay. For that purpose, we used SCR values of healthy controls and SMA patients.

The SCR value of healthy controls was 0.59 ± 0.12 (mean ± SD), whereas the SCR value of SMA patients was 0.13 ± 0.01. The SCR value differences between the two groups were statistically significant (*p* < 0.01) (Figure 4).

The SCR value ranges of controls and SMA patients were far apart and did not overlap in this study. An SCR value of 0.2, however, was tentatively set as the cutoff point showing the absence of *SMN1*. This tentative cutoff point was less than −3 SD of healthy control values, 0.23.

## 4. Discussion

### 4.1. Expansion of SMA Screening to Older Children or Adults

There may be quite a few undiagnosed cases of SMA in all age groups. The NBS-SMA is a nationwide public health program intended to identify newborn infants with SMA. However, the number of newborns screened for SMA is too small to detect all affected infants with SMA [20]. A nationwide spread of the NBS-SMA program may not happen soon. In addition, infants after the neonatal period will not be tested in the NBS-SMA. Diagnosis of SMA may become more difficult after the neonatal period. Milder symptoms can make a diagnosis of SMA more difficult because of similarities with other muscle weakness diseases [24,30].

To increase the diagnosis rate, we must consider adding SMA screening into the health check items for infants, toddlers, school-age children, and adults. Our system may be the best option for SMA screening as part of the health checks. As a non-invasive and cost-effective alternative, DSS sampling is suitable for health checks held in schools or workplaces. Once dried, DSS samples can be stored at room temperature in the dark, without any special equipment, and they can be sent to a center or a laboratory by airmail, enabling telediagnosis and eliminating hospital visitation.

### 4.2. Saliva as a Good Source for Genetic Analysis

Saliva has been used for analysis of human DNA by PCR-based HLA typing [31], microarray SNP genotyping [32,33], TaqMan SNP genotyping assays [32,34], loop-mediated isothermal amplification-melting curve (LAMP-MC) analysis [34], PCR-Sanger sequencing [33], next generation sequencing [35], and whole genome or exome sequencing [36,37].

Saliva has now proved to be a good alternative source for genetic analysis. In the studies mentioned above, saliva was collected using a saliva collection kit with a special container, either in-house-made [31] or commercially available [32,33,34,35,36,37]. Later, DNA was extracted from the liquid saliva sample in the container [31,32,33,34,35,36,37].

Some previous studies used a different kind of saliva sample. Instead of a liquid saliva sample, some researchers prepared DSS samples and extracted DNA from them for various purposes [38,39,40]. Kisoi et al. identified insertion/deletion polymorphisms of the angiotensin-converting enzyme (*ACE*) gene by PCR amplification using DNA from DSS samples [38]. Our study also demonstrated genotyping analysis of *SMN* genes using DSS samples. These findings support the idea that saliva, even if dried, can be a good DNA source for PCR.

Kisoi et al. reported a simple DNA extraction method from the DSS samples: they boiled one DSS punch (4 mm in diameter) and used an aliquot of the supernatant for PCR amplification [38]. Hayashida et al. reported direct PCR from DSS to detect SNP in alcohol metabolism-related genes (*ADH1B* and *ALDH2)* using a TaqMan probe assay [41]. In agreement with their results, we provide further evidence that direct PCR amplification is feasible from DSS without boiling or DNA purification procedures.

### 4.3. Robustness and Accuracy of Our SMA Screening System

In this study, the basic technology to detect *SMN1*, mCOP-PCR, is a type of allele-specific PCR. Originally, COP-PCR was based on allele-specific amplification, in which two oligonucleotide primers compete for one target DNA sequence in one PCR tube [42]. The primers are short (10–11 bases) and highly identical except for one nucleotide change in the middle of the primers [8,42]. The modified version adopted in this study, mCOP-PCR, uses one gene-specific oligonucleotide primer (SMN1-COP) that preferentially binds the *SMN1* sequence rather than the competitor sequence, *SMN2* [8].

Here, we employed nested PCR to rescue low quantity and quality genetic material in the DSS. The first-round PCR amplified common regions of the target genes and provided enough template for the second-round PCR, even for quantitative assays [7,8,43]. In addition, the first-round PCR products contained no other sequences similar to the *SMN1*-specific mCOP-PCR primer, which avoids the amplification of unexpected primer binding sites in the second-round PCR and enhances the specificity of our system [7,8].

### 4.4. Limitation of PCR-Based Assays Using DSS

The use of saliva, either liquid or dried, has potential limitations for PCR-based assays. First, the quantity of amplifiable human DNA recovered from liquid saliva is lower than blood (37.3% in saliva versus 87.6% in the blood) [32]. Second, the DNA quality obtained from saliva does not always meet the quality standard for certain assays, even though the DNA obtained from saliva showed a good A260/A280 ratio [36]. DNA degradation during the storage period may be related to the quality. Third, PCR inhibitors in the saliva may confound the results [32,44].

We experienced one case of assay failure out of 61 DSS samples (1.6%) in this study. This failure might be due to the absence of DNA or the presence of PCR inhibitors. The possibility of DNA degradation can be ruled out because the storage period was less than two weeks. In our pilot study of NBS-SMA, there were no cases of assay failure out of 4157 DBS samples (0%) [7]. Compared with our previous study of DBS, this study with DSS showed a higher rate of assay failure.

## 5. Conclusions

We demonstrated the potential use of DSS sampling as a good alternative source for SMA detection using nested mCOP-PCR. The sample collection procedure is non-invasive, easy to handle, and requires no hospital visitation. DSS might be preferable for SMA screening for older children or adults. We hope that this will help in overcoming the delay in SMA diagnosis.

## Figures and Tables

**Figure 1 genes-12-01621-f001:**
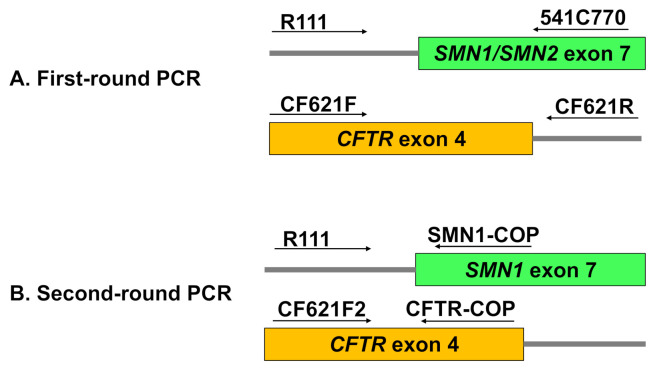
Primer locations for (**A**) the first-round PCR and (**B**) the second-round PCR. Arrows indicate the direction of the primers.

**Figure 2 genes-12-01621-f002:**
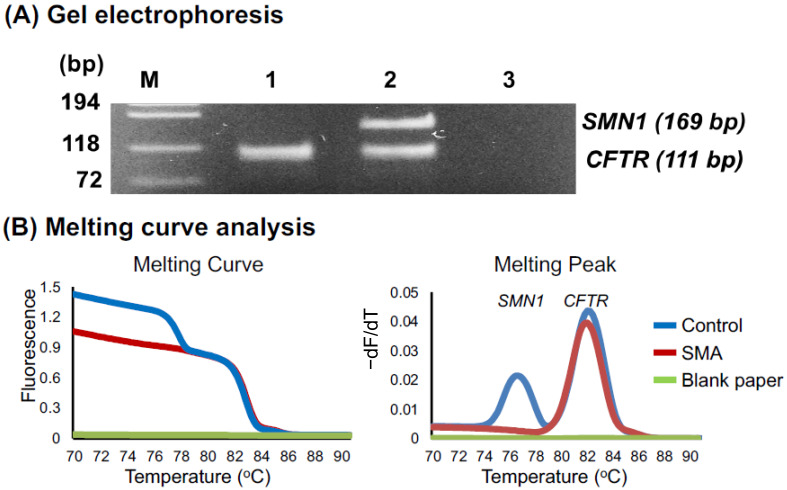
Detection of *SMN1*. (**A**) Agarose gel electrophoresis of the second-round PCR products indicating the *SMN1* band (169 bp) and the *CFTR* band (111 bp). Lanes M, 1, 2, and 3 represent the DNA ladder, SMA patient, healthy control, and blank paper, respectively. (**B**) Melting curve and melting peak analysis to detect *SMN1*/*CFTR* in a healthy DSS (blue line), an SMA DSS (red line), and a blank paper (green line).

**Figure 3 genes-12-01621-f003:**
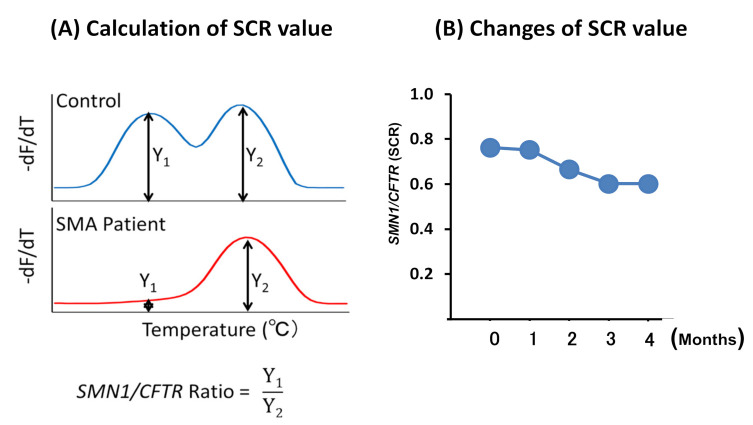
The *SMN1*/*CFTR* peak ratio (SCR) value. (**A**) The formula to calculate SCR. (**B**) The changes to the SCR value over the month(s).

**Figure 4 genes-12-01621-f004:**
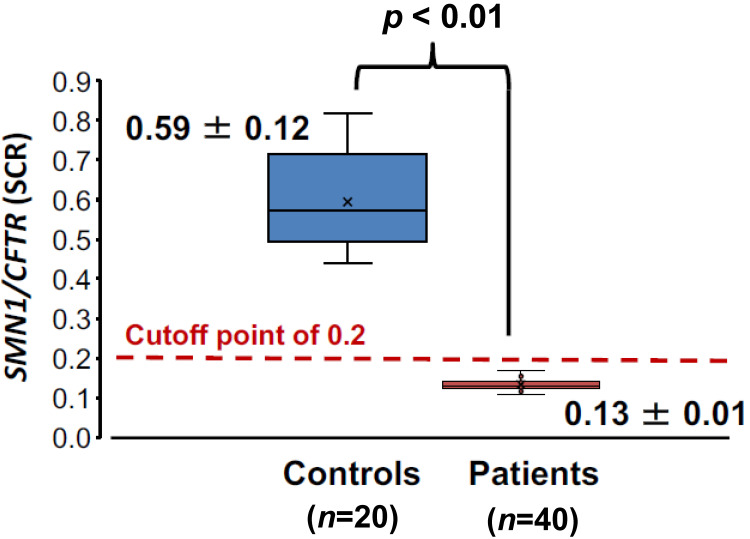
Box-and-whisker plot of the SCR values from controls (blue box) and SMA patients (red box). The “x” mark represents the average value. The middle, lower, and upper lines of the box represent the median value, the 25th percentile, and the 75th percentile, respectively.

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
