# Peer review of "Detection of Spinal Muscular Atrophy Patients Using Dried Saliva Spots"

_genes, 2021, doi:10.3390/genes12101621_

Round 1
Reviewer 1 Report
- The aim of this study for development of a non-invasive SMA screening method with DSS was clear and straight forward. This was an application of DSS, targeting to replace DBS with DSS for screening.
- The research design was applied from a previous study with DBS for screening, which is not original. However, this paper presented details of controls for detection and calculation of PCR results, which was supportive for the research design.
- For validation and detection, please add the rationale of setting up the cutoff point as 0.2 for SCR value.
Reviewer 2 Report
The authors present an interesting technical report describing the detection of SMN1 deletion in genomic DNA samples derived from dried saliva spots (DSSs). While the relevance of using a minimally invasive specimen for genetic testing is high, the detection method for SMN1 is not very sensitive. Also, the rationale for using CFTR as a comparator is not provided.
There would greater enthusiasm for this approach if SMN1 and SMN2 copy numbers could be reliably measured from DSS-derived DNA.
Additionally, the authors need to describe more recent techniques for SMN1 deletion detection, like those based on digital droplet PCR and on next generation sequencing.
Round 2
Reviewer 2 Report
The concerns raised regarding this technical report have been sufficiently addressed; however, the limitations of using a comparator gene that has been linked to a human genetic disorder is not addressed. Also, the authors seem to be selective in their citations of recent key references of other methodologies.